# 'I actually felt like I was a researcher myself.' On involving children in the analysis of qualitative paediatric research in the Netherlands

Malou L Luchtenberg  ,[1,2] Els L M Maeckelberghe,[3,4] AA Eduard Verhagen[1,2]

[1]University of Groningen, Groningen, The Netherlands
[2]Beatrix Children's Hospital, University Medical Center Groningen, Groningen, The Netherlands
[3]Institute for Medical Education, University of Groningen, Groningen, The Netherlands
[4]University Medical Center Groningen, Groningen, The Netherlands

**Correspondence to**
Malou L Luchtenberg;
m.l.luchtenberg@umcg.nl

## ABSTRACT

**Objectives** To evaluate the feasibility of a new approach to paediatric research whereby we involved children in analysing qualitative data, and to reflect on the involvement process.

**Setting** This was a single-centre, qualitative study in the Netherlands. It consisted of research meetings with individual children at home (Phase I) or group meetings at school (Phase II). In Phase I, we identified themes from a video interview during five one-on-one meetings between a child co-researcher and the adult researcher. In Phase II, during two group meetings, we explored the themes in detail using fragments from 16 interviews.

**Participants** We involved 14 school children (aged 10 to 14 years) as co-researchers to analyse children's interviews about their experience while participating in medical research. Notes were taken, and children provided feedback. A thematic analysis was performed using a framework approach.

**Results** All co-researchers identified themes. The time needed to complete the task varied, as did the extent to which the meetings needed to be structured to improve concentration. The children rated time investment as adequate and they considered acting as co-researcher interesting and fun, adding that they had learnt new skills and gained new knowledge. The experience also led them to reflect on health matters in their own lives. The adult researchers considered the process relatively time intensive, but the project did result in a more critical assessment of their own work.

**Conclusion** The new, two-phase approach of involving children to help analyse qualitative data is a feasible research method. The novelty lies in involving children to help identify themes from original interview data, thereby limiting preselection of data by adults, before exploring these themes in detail. Videos make it easier for children to understand the data and to empathise with the interviewees, and limits time investment.

## INTRODUCTION

Researchers should be wary of interpretation bias when analysing data of qualitative studies. Qualitative research includes a subjective component that calls for reflexivity.[1] Because the life experiences and social situations of children differ from those of adults, their

### Strengths and limitations of this study

► This study describes a new approach to paediatric research whereby children are involved in analysing qualitative interview data.
► The novelty of this study lies in the fact that children are involved in helping to identify themes from original data as well as exploring the themes in more detail.
► This study explores the use of videos rather than transcripts to present the interviews to relatively young co-researchers.
► The study reflects on children's involvement as co-researchers from the perspective of the children themselves and from that of the adult researchers.
► A limitation of this study is that, in the test phase presented here, we limited the number of child co-researchers and selected the interviews from a larger data set to include as much variation as possible.

interpretations of data derived from interviews with children may differ from adults' interpretations. It is therefore desirable to involve children to strengthen the analysis of such qualitative data. There is little evidence in the literature on how children could be effectively involved in scientific data analysis.[2–9] Other challenges regarding patient and public involvement (PPI)[10] of children include lack of funding and time, gatekeeping or power imbalances, and concerns about obtaining knowledge and training on how to involve children.[2 5 11–17] Measuring the impact of a PPI process on research output is difficult because the involvement process itself is complex and therefore its impact cannot by fully captured by evaluating outcomes.[18]

Best and colleagues recently introduced a new method called participatory theme elicitation, whereby a youth advisory panel is involved in qualitative data analysis. The method involves capacity building (training), data selection by adult researchers, data sorting by youth members, and final grouping

and analysis by adult researchers.[19] A disadvantage of the method is that it involves data preselection by adults. We hypothesise that it is feasible to involve children from the beginning of data analysis, starting with the identification of themes from original data.

In a larger unpublished study, we collected the experiences of young people regarding their participation in medical research in order to provide recommendations for improving children's participation in research. In the present study, we aimed to explore whether it is feasible to involve children in the analysis of the qualitative data. We designed a two-phase approach that would be effective and efficient: effective in the sense that it involved children to identify themes as well as to explore the themes in more detail, and efficient in the sense of limiting time investment. We also aimed to reflect on the involvement process from the perspective of both adults and children. The results of the qualitative data analysis are to be published elsewhere.

## METHODS

This was a single-centre study performed by researchers of the University Medical Center Groningen, the Netherlands. We investigated the involvement of children as co-researchers in identifying and analysing themes from interview data presented on video. We conducted the investigation in two phases: Phase I consisted of five individual meetings, and Phase II of two group meetings. In addition, we reflected on the children's involvement in the analysis process.

### Recruitment and sampling
#### Phase I
We approached potential co-researchers through national patient support organisations, primary schools, hospitals, social media and by word-of-mouth. No research experience was required of the participants. Sampling was based on age (9 to 18 years) and participants were required to be fluent in Dutch because the meetings were held in that language. Initially, seven children volunteered. They each received an information leaflet. Even though recruitment was challenging, we managed to achieve our goal of five participants in Phase I.

#### Phase II
We recruited the participants for Phase II in collaboration with a specific primary school in Groningen, the Netherlands, where teachers and learners are expected to display an academic mindset aimed at research and analysis.[20] We invited one class of 15 children of the school's oldest learners to participate. Ten learners volunteered, one of whom was unable to participate because of illness.

### Informed consent
#### Phase I
We supplied the potential participants with verbal and written information. We asked them to discuss the study

with their parents and to reply by post, e-mail or telephone. Before the session at the child's home started, he or she read the informed consent form and discussed it with the researcher and a parent. One parent was present throughout the session, but was kindly asked to not interfere with the process. All the children, irrespective of age, were asked to sign the informed consent form to acknowledge that we appreciated their contribution equally. In accordance with the Dutch Medical Research Involving Human Subjects Act, parental consent was obtained in addition to the child's consent.[21] Children were also asked to sign a confidentiality agreement regarding any personal information present in the data they analysed. At the end of the session, we gave the participants a €10 gift voucher as token of our appreciation of their time and a certificate acknowledging their contribution as co-researchers.

#### Phase II
Two adult researchers visited the primary school to meet the learners, their teacher and headmaster, and to introduce the research project. The headmaster agreed to allow the children to participate during school hours. The potential participants received an information leaflet similar to the one given to the participants of Phase I. We asked them to complete the consent form and the confidentiality agreement at home with a parent, and to return the form to their teacher. At the end of the group meetings, we gave the participants a certificate. These children were not given a gift voucher because they participated during school hours.

### Data characteristics
Data from the original interview study were analysed by our co-researchers. In addition, the adult researchers took notes during the analyses and written feedback was obtained from the participants before, during and after participating. In table 1, we provide additional information on the data collected for the original interview study. Out of a total of 23 interviews, two were excluded because they were not videotaped. Another five were excluded because participants and parents had not consented to use the video data.

The co-researchers received a brief interactive introduction to paediatric research and to the original interview study. They were asked to identify the main topics in video interviews and to summarise them on mind maps ("a type of diagram with lines and circles for organising information so that it is easier to use or remember").[22] To prevent the children from becoming 'little adult researchers' we did not train them extensively. Extensive training would also have been more time consuming for both the children and the researchers.

In Phase I the five co-researchers collaborated with the coordinating researcher (PPI adult researcher (PPIA) 1) in a one-on-one session to identify the main themes in five different interviews. Together, they watched a video of between 25 and 45 min of another young person and

**Table 1** Details of data of the original interview study on children's experiences in medical research

| Study characteristics | |
| --- | --- |
| Aim | To explore children's experiences in medical research to obtain recommendations from their perspectives on how to improve children's involvement in research. |
| Setting and research team | Single-centre study conducted by a team of researchers at University Medical Center Groningen, the Netherlands. The research team consisted of an ethicist (EM), paediatrician (EV) and MD/PhD student (ML). All members were trained researchers and/or had previous experience in conducting qualitative research. |
| Recruitment and sampling | Recruitment through health providers from several hospitals, national patient support groups, social media and by word-of-mouth. Purposive maximum variation sample: children, patients as well as healthy volunteers between 9 and 18 years old who were invited to participate in different types of medical research in the Netherlands and who either took part or declined to take part. The participants had no prior relationships with the members of the research team. |
| Informed consent | Informed consent given by one parent and the child or, in accordance with Dutch law, from 16 years and older by the child only. |
| Data collection | Twenty-three semi-structured, in-depth interviews, lasting between 30 and 100 min, with children about their experiences in taking part in medical research, including recommendations for improvement of children's involvement in informed consent procedures and the research itself. A topic guide was developed based on a previous study in the UK. Interviews performed by ML took place at children's homes and were recorded on audio or video, transcribed verbatim and returned to the participants. No comments from participants were received. Data collection continued until we reached data saturation of main themes. |
| Ethical approval | The conclusion of the Medical Ethics Review Board of the University Medical Center Groningen was that this study, no. M16.192386, 10 May 2016, fell beyond the scope of the Dutch Medical Research Involving Human Subjects Act. |

discussed the emerging themes. This took place at the participants' home and lasted between 2 and 3.5 hours. During Phase I, the adult researcher spent a total of 11 hours travelling to and from co-researchers homes.

During the two group meetings in Phase II, we explored in detail two themes that had been identified during Phase I. For this purpose, we compiled two 5 min videos from fragments of several interviews from the data set of the original study. The group meetings took place at a local primary school and lasted approximately 2.5 hours. Travelling time for adult researchers was 2 hours, including the introductory meeting when we handed out the information sheets. In addition to travelling time, the time investment of adult researchers was approximately 5 hours. This included the time spent preparing the instructions and making the compilation videos. The material costs of the project were low and the travelling costs minimal.

The aim of the analysis process in both phases was to identify the main themes in the video through open, unstructured discussions with the child co-researchers. To facilitate interaction and discussion, the co-researchers and the adult researchers could pause the video at any time. They took notes of what they thought the interviewee found important. PPIA1 and PPIA2 allowed moments of personal reflection, but were on guard for potential intertwining of co-researchers' personal experiences with interviewees' experiences. After watching the video, the co-researchers drew a mind map depicting their interpretation of the connection between different themes.[22] The mind maps were drawn on A3-sized sheets of paper, using

different sizes of sticky notes, and coloured pens. Besides, the participants were free to use materials of their own choice. The researcher asked the participants questions about the importance of certain themes, the identification of overlapping themes and the reason why they had chosen a certain theme. The participants led the discussion and made the final decision in case of a disagreement about a theme.

### Data collection and analysis

Before the study actually commenced, we asked the child co-researchers why they would like to take part. The adult researchers took notes of how participants fulfilled their role as co-researcher, and how the participants reflected on this process. Child co-researchers completed a feedback form after the analysis (table 2). In addition, we briefly evaluated the process orally. All meetings were recorded on audio tape.

We performed a thematic analysis using a framework approach.[23] Familiarisation and initial theme identification was done by PPIA1 (ML) and discussed with EM (individual meetings), and PPIA2 (group meetings) based on the audio tape, notes and the participants' written feedback. Some themes, such as time investment, were identified in advance from the literature, others were derived from the data. Themes were refined and conceptualised during regular meetings with the research team and any disagreements were discussed and a final decision reached by consensus. Because the sample for this exploratory study was relatively small, we did not aim for data saturation.

**Table 2**  Feedback form

| | |
|---|---|
| 1. Did you understand beforehand what your role was in the project? (No / a little / yes) | 6. Would you recommend other children to become co-researchers? Why? |
| 2. How could we improve the information about working as a co-researcher? | 7. How would you rate your time investment? (Too long, adequate, too short) |
| 3. What was it like for you to work as a co-researcher? | 8. a. What did you think of the €10 gift voucher? (Phase I) |
| a. Positive aspects | b. What did you think about participating in this project during school hours? Why? (Phase II) |
| b. Points of improvement | 9. Do you have suggestions for improving this evaluation form? |
| 4. Did you learn anything from being a co-researcher? If so, what did you learn? | |
| 5. Would you like to be a co-researcher more often? | |

## Patient and public involvement

We explored the feasibility of an approach whereby children were involved in qualitative data analysis, both in helping adult researchers to identify themes from original data as in exploring the themes in more detail.

## RESULTS

### Participant characteristics

Fourteen children, eight girls and six boys, participated as co-researchers in this study. Two participants had experience because they had been chronic patients themselves. None of the children had been a co-researcher before. Tables 3 and 4 show the characteristics of the child and adult participants.

### Reflection and evaluation of the involvement process

The results can be divided into five main themes: (1) understanding the study procedures, (2) empowerment, (3) reflection on health and illness, (4) interest in the bigger picture, and (5) reflection on time investment. The results of the feedback form are summarised in table 5.

#### Understanding of study procedures

When we asked the participants in the group meetings what they expected of the session, they remembered that the main idea was 'doing research about research'. Nonetheless, not everyone remembered the details such as whether photographs would be taken. Even though the information was especially written with children in mind, the participants recalled that it was mostly read and signed by their parents, and had not always been discussed with them. In the case of the co-researchers in the individual meetings, this was different. Here the researcher and the parents were present and encouraged the children to complete the form themselves. Parents were available in the background in case their help was needed.

Most children reported that they had a general idea about what their role was in the project. We explained their role to them in detail during the actual project. One of them explained:

'I already understood it but once you are doing it, you understand it (better).' (Girl, individual meeting)

#### Empowerment

By involving children as co-researchers, they gained knowledge, learnt new skills and became more confident

**Table 3**  Child participant characteristics

| Characteristics of child co-researchers | N* (%)† |
|---|---|
| **Sex** | |
| Girl | 8 (57) |
| Boy | 6 (43) |
| **Age (years)** | |
| 10 | 1 (7) |
| 11 | 10 (71) |
| 12 | 2 (14) |
| 13 | 0 (0) |
| 14 | 1 (7) |
| **School attended** | |
| Primary school | 13 (93) |
| Secondary school | 1 (7) |
| **Hospital/disease experience (lived experience)** | |
| Currently suffering from a disease | 2 (14) |
| Hospitalisation or minor surgery in the past | 6 (43) |
| Family member(s) who suffer from a disease | 2 (14) |
| None | 4 (29) |
| **Research experience as participant** | |
| Yes | 2 (14) |
| No | 12 (86) |
| **Experience as co-researcher** | |
| Yes | 0 (0) |
| No | 14 (100) |

*Number of child participants; n=14.
†Rounded to nearest whole number.

**Table 4** Adult participants characteristics

| Participant | Sex | Experience illness/hospital | Research experience (participating in research) | Research experience (performing research) |
|---|---|---|---|---|
| PPIA1 | Woman | As a medical student. Graduated as medical doctor in August 2019. | Yes. Participant in two large cohort studies for several years. | Training and experience in qualitative research for PhD. |
| PPIA2 | Woman | As a medical student. Started internships in September 2019. | No previous experience in medical research participation. | Trained in qualitative research as a former psychology student. |

PPIA, patient and public involvement adult researcher.

in fulfilling their role during the analysis process. This was mostly because they enjoyed the new experience of contributing to research or helping others. Children felt empowered by working as a co-researcher. One of them explained it as:

'It actually felt a bit like I was a researcher myself.' (Girl, group meeting)

They reported gaining new knowledge about certain health matters and they realised what it is like to be ill. They learnt how to do research, how to think critically and how to take notes. Here are some representative answers given:

'You have to think carefully before you draw conclusions.' (Boy, group meeting)

'A bit about how ill children felt afterwards (after participating in research).' (Girl, group meeting)

All the participants were positive about the idea of being a co-researcher more often, mostly for similar reasons as for wanting to take part in the first place:

'Yes, it was fun, relaxed and instructive.' (Girl, group meeting)

One participant, the only one attending secondary school, mentioned it would depend on whether he had the time to take part because of homework and sports activities in his free time. All participants reported that they would recommend others to become co-researchers, though one participant acknowledged that it might not suit everyone. She mentioned that some children might not enjoy it or might not have the skills to do such work.

During the analysis process in Phase I, the co-researchers grew noticeably more confident as time progressed. The adult researcher retreated to the background and stimulated the co-researchers to take the lead, which most of them did eventually. One participants actually asked the researcher questions about her observations, instead of the other way around. Most participants, however, needed some form of structuring support from the adult researcher throughout.

During the group meetings the support needed from the adult researcher was different. The co-researchers needed more structuring because of group dynamics. Confusion due to competing voices ensued, and shy children tended to not be heard. Interestingly, the two groups chose different ways of translating their notes to the mind map. In one group, the adult researcher noticed a clear distinction in participants' role preferences, and the co-researchers divided the roles between themselves. Some preferred an executive role, such as writing down themes on the 'sticky notes', while others preferred to simply express their ideas and play a more coordinating role. Some found it difficult to summarise their notes and suggested first underlining important notes:

'We could also just first underline what we think is important.' (Girl, group meeting)

The co-researchers in the other group together decided that they all wanted to write down their own notes on 'sticky notes' and to put them all on the mind map. The outcome was a mind map that displayed different topics as well as providing insight into how important the individuals thought a certain topic was by the number of 'sticky notes' on the same topic. Others only needed a bit more time and space to find their own role.

It was challenging for the adult researchers to not provide answers themselves when the participants indicated that they did not know how to proceed with the analysis. By repeating or rephrasing their question and by acknowledging that they were doing the right thing, the adult researcher could reinforce the children. Both adult researchers were surprised by the co-researcher's achievements. Throughout the project, the co-researchers displayed the ability to identify themes and to visualise them in mind maps, underlining the feasibility of this approach and its value for interpreting data.

The researchers learnt a great deal from involving children in the analyses because the participants were very open and direct in their feedback. If they did not understand something, for example, if a question was not clearly formulated, or if they did not understand medical jargon such as 'treatment protocol', they said so immediately.

Whereas the adult researchers tended to generalise findings, the co-researchers stuck more to the original data. One topic, for example, dealt with recommendations to researchers for improving young people's experiences when participation in medical research. The interviewees mentioned issues like making hospital visits more enjoyable and gave concrete suggestions. All the researchers, adults and children alike, started from the original data, but there appeared to be a difference in analysis. The

**Table 5** Summary of written feedback

| Theme (question from (table 2)) | Summary of written feedback | N* (%)† |
|---|---|---|
| Understanding the study procedures (1,2,9) | Understanding their role as co-researcher before start‡ | |
| | No | 0 (0) |
| | A little | 12 (86) |
| | Yes | 2 (14) |
| | Suggestions for improving the information about working as co-researcher | |
| | Don't know | 4 (29) |
| | Everything was clear | 5 (36) |
| | Use fewer difficult words | 4 (29) |
| | Explain that we had to take notes and create a mind map | 1 (7) |
| | Suggestions for improving the feedback form of the co-researcher project | |
| | Adding a question about the overall experience | 1 (7) |
| | No recommendations | 13 (93) |
| Empowerment (3 to 6) | Positive experience as co-researcher | |
| | Fun | 14 (100) |
| | Interesting | 4 (29) |
| | Helping other children | 1 (7) |
| | Learning something new | 1 (7) |
| | Time investment was okay | 2 (14) |
| | Receiving a certificate | 1 (7) |
| | Points of improvement for co-researcher project | |
| | No points of improvement | 12 (86) |
| | Shorter interviews | 1 (7) |
| | The project should take the whole school day (instead of a half day) | 1 (7) |
| | Lessons learnt from being a co-researcher | |
| | Taking notes | 2 (14) |
| | Critical thinking and listening | 4 (29) |
| | About a medical condition | 2 (14) |
| | About doing research | 2 (14) |
| | About how children think and feel about research | 2 (14) |
| | That children think differently from adults | 1 (7) |
| | That it is fun and that you learn a lot | 1 (7) |
| | Not really | 1 (7) |
| | Would like to be co-researcher more often including reason | |
| | Yes, because it's fun | 12 (86) |
| | Yes, because it's interesting | 5 (36) |
| | Yes, because I like to help people | 1 (7) |
| | Yes, I know what to expect now | 1 (7) |
| | Yes, if it doesn't hurt | 1 (7) |
| | It is fun, but depends on how much time I have | 1 (7) |
| | Would you recommend others to become co-researcher? | |
| | Yes, because it's (super) fun | 9 (64) |
| | Yes, because it's interesting/you learn something from it | 7 (50) |
| | Yes, because you receive a gift voucher | 1 (7) |
| | Yes, because you can help other people | 2 (14) |
| | Yes, because you get sweets | 1 (7) |
| | Yes, but it depends on whether it suits them | 1 (7) |

**Table 5** Continued

| Theme (question from (table 2)) | Summary of written feedback | N* (%)† |
|---|---|---|
| Time investment (7,8b) | Rating of time investment‡ | |
| | Too long | 1 (7) |
| | Adequate | 12 (86) |
| | Too short | 1 (7) |
| | Thoughts on having this project during school time (Phase II, n=9)§ | |
| | Fun/good, because you didn't have to do schoolwork | 6 (67) |
| | Fun/good, because you don't miss free time after school | 4 (44) |
| | Don't mind | 1 (11) |
| Compensation (8b) | Thoughts on receiving a gift voucher (Phase I, n=5)§ | |
| | Fun/good | 5 (100) |
| | Not necessary | 2 (40) |
| | Creative | 1 (20) |

*Number of child participants; n=14. Some participants provided more than one answer.
†Rounded to nearest whole number.
‡The feedback provided is based on a multiple choice question.
§Calculation based on a sub-selection of total participants, because children took part in a different phase.

concrete suggestions in the children's analyses remained and their recommendations were therefore emphasised, while the adults generalised them into 'ways to brighten up the visits'. The co-researchers brought the adults back to the basics that were important to children.

### Reflection on health and illness
The participants empathised with the young people who shared their experiences in the video. They wondered whether they were still ill or asked if they still lived and hoped they were all right. The following is a representative example:

'None of these children (in the video) is deadly ill, right?' (Girl, group meeting)

The adult researcher explained the different illnesses the interviewees had, and mentioned that some of them had been critically ill, for example, with leukaemia, but that they were stable at the time they were interviewed.

The co-researchers asked questions about the illnesses and what the consequences might be for the lives of the children such as two siblings with a hereditary condition. They also shared their own experiences, for example, about relatives who had cancer.

### Interest in the bigger picture
Even though we only involved the participants in the analysis stage of the main interview study, they were well aware that this was part of a bigger study. They asked about the other participants and previous experiences with involving children in research analysis, and wondered whether we would show the mind maps they had made to the interviewees.

Some of the questions they asked were:

'Have you been to children's homes?' 'Are you also doing this (project) at other schools?' 'How many times have you done this?' (Several co-researchers,

boys and girls, group meetings, not clearly identifiable from the audio)

Hearing that they were the first group of children that participated as co-researchers in this project made them feel special. They expressed the wish to receive the final results and hoped we would do this project again. The co-researchers had a broader interest than just fulfilling their role as co-researchers. They also asked personal questions, such as why the adult researchers did this research project and whether it was part of their university training. They also acknowledged and enjoyed helping the adult researchers with their research:

'It is, of course, good for you (adult researchers) that we participate so that you can continue doing research about research of the research.' (Girl, group meeting)

### Reflection on time investment
Most children said the time investment was appropriate. One child in the group meetings reported that a shorter meeting would have been better because some children became distracted:

'Because at the end we were chatting a bit, (we got) distracted.' (Boy, group meeting)

This was also observed by the adult researchers. Another participant, however, reported that he would have like the session to last longer because he really liked it and did not want to return to his normal schoolwork. All the Phase II participants reported that it was an interesting and fun alternative to normal school tasks. They thought it was good to do the project during school time because this way they would not miss out on any free time. One of the participants explained this:

'It was fun because you didn't have to work, and if it hadn't been during school, it was inconvenient.' (Girl, group meeting)

From the adult perspective, time was invested in recruitment, developing material to introduce and explain the procedure, and thinking about how to best involve children. The actual analysis with co-researchers was time-intensive and lasted longer than we had expected. Nevertheless, the time invested was considered reasonable given the empowerment of children, their learning new skills and their views on our data provided additional insight, which is promising for the interpretation of our interviews.

## DISCUSSION

Little evidence is available on how to involve children in research.[11] In this paper, we describe how we involved relatively young children in the analysis of medical research interviews analysis using a two-phase approach. We deployed various strategies to avoid a tokenistic approach, to address challenges regarding time management and to empower children during the process.

### Two-phase approach

Involving relatively young children, aged 10 to 14, in our interview analyses was a challenging process. Our aim was to limit preselection of data by adults by introducing one-on-one meetings during which research interviews were analysed by young co-researchers. This approach could be considered an extension of Best's 'participatory theme elicitation' method.[19] Even though the sessions lasted longer than originally intended, the fact that we could focus on one individual worked to our advantage. In our opinion, the project benefited from the fact that the individual meetings were held at the children's homes, which constituted a safe and familiar environment. This confirms findings from Dovey-Pearce and colleagues who highlighted the importance of face-to-face meetings to establish relationships.[18] The themes identified during Phase I were explored in detail during group meetings in Phase II. Working with a number of co-researchers in this phase improved the rigour of the qualitative analysis. In addition, there was an unexpected positive result for the co-researchers who were classmates. Their bond was strengthened by their shared research experience and by reflecting on health and illness together. The two-phase approach enabled us to achieve our research goals and to empower our co-researchers, while keeping within reasonable the time limits. This applied to adult and child researchers alike.

### Use of videos in analysis

Our aim was to involve child co-researchers in interview analyses in an effective and efficient way. Data analysis in qualitative research is often a lengthy process involving large quantities of text. Locock and colleagues reported that young people reading through the transcripts was tedious and inefficient. They concluded that it was more effective to discuss the data rather than digging into detailed transcripts.[9] We decided to explore other ways of involving young co-researchers in interview analyses. Visuals such as photographs, drawings or mapping methods are often used in collaboration with young children to collect data about children's views on, for example, what they value in their lives.[24] Darbyshire and colleagues reported that using a variety of qualitative visual techniques is helpful for engaging children in research. It also provides a good way for children to express their views. There is, however, a problem with using visuals as a participatory method rather than in analysis, as Darbyshire and colleagues pointed out: "…having children take photographs and then having only adults 'interpret' (or possibly misinterpret) them is potentially an adultist approach to research on children that we sought to avoid."[25] For this reason, we used videos in the analysis stage to visualise the interview data to be analysed. Our study confirmed the benefits expressed by Darbyshire and colleagues. The co-researchers enjoyed the creative process of developing the mind map and the videos helped them understand and empathise with the interviewees. Using videos rather than transcripts made the process more time-efficient, while preserving the effectiveness of a thematic analysis. Another benefit of videos over transcripts concerned the rigour of the qualitative data analyses. The analysis of interview data is often assumed to start as soon as the interview has been fully transcribed but, even in case of a verbatim transcript including descriptions of vocal emotions such as laughter, the loss of key elements, such as volume of voices and facial expressions, remains. This could present interviewees' experiences in a more abstract way than the original data show.[26 27] Put differently, by using videos, we possibly started the analysis with a more authentic representation of the data.

### Additional considerations and further research

Time investment is an important consideration when developing ways of involving children in research analysis. In the method developed by Best and colleagues, the analysis is limited to 2 hours. An additional time investment of four times, 90 to 120 min, is asked of participants for 'capacity building.' In these sessions, young people learn how to design and conduct a study, how to perform qualitative data analyses, and they receive an introduction into the subject matter of their data.[19] We purposely did not train our co-researchers to avoid shaping them to comply with our idea of a qualitative researcher, and to limit time investment. Though we cannot make a comparison, the little training we gave our co-researchers did not seem to have had a negative impact on the result.

In addition to time investment, timing of research meetings should be considered. Many Young People's Advisory Groups generally plan their meetings during school holidays or weekends.[28] The INVOLVE Advisory Group of the UK's National Institute for Health Research, which supports active public involvement in the National

Health Service, public health and social care research,[29] identified parents and schools as a significant barrier to public involvement during school hours: "…lack of schools' recognition of the value of their work sometimes acts as a barrier to them attending events which involve travel in school hours."[13] Nevertheless, thanks to a cooperative headmaster, we managed to set up a collaboration with a primary school for Phase II of our study, and planned the group meetings during school hours. For our co-researchers in the group meetings, participating during school time was preferable to after school, because they reported that they had busy schedules or had to do homework during their free time.

Our results showed that children tend to include more concrete topics in their analyses, whereas adults analyse data in a more abstract way. This is in line with the cognitive development of children, who transform from concrete to abstract conceptualisation later in adolescence.[30 31] Consequently, we expect that an evaluation of the data analysis process performed by children, young people and adults will provide additional interesting insights. Recently, we started testing our two-phase approach with young people aged 16 to 18 years. They will use this project for a school assignment, thereby creating a situation that is mutually beneficial. If this proves successful, we will consider setting up a long-term collaboration with primary and secondary schools to optimise collaboration between researchers and children to help decrease the knowledge gap between academia and society.

## Study limitations

To test this new approach, we started with a small group of young co-researchers. As a consequence, we had to select of interviews from a larger data set to analyse in Phase I. We aimed for as much variation as possible within this selection, but we also needed to be pragmatic, regarding the length of the interviews for instance. In addition, in our method children were not involved in making choices about specific quotes used in the results sections. As recruiting co-researchers was challenging, sampling was limited to age and to fluency in the Dutch language. Maximum variation sampling would, however, be preferable to improve reflexivity.

## CONCLUSION

We suggest that the two-phase approach to involving young children in analysing qualitative data is feasible. Its novelty lies in recruiting children to help identify themes from original data before the themes are explored in detail. Thus, preselection of data by adults is limited. By combining one-on-one meetings and group meetings, the two-phase approach is an effective and efficient way of involving relatively young children in analysing qualitative data. Additional benefits are that children reflect on health and illness in their own lives, they are empowered and engaged in medical research. We recommend presenting the interview data on videos rather than through transcripts. Videos make it easier for children to understand the data and to empathise with the interviewees, and it limits time investment.

**Acknowledgements** We thank all the participants in the original interview study and our co-researcher participants in this project. Because this is a meta-analysis of the patient and public involvement (PPI) project, we decided not to invite the co-researchers as co-author of this article. We thank Laura Postma for helping to facilitate the group meetings.

**Contributors** ML, EM and EV designed the original study. ML was responsible for the data collection in Phase I and the group meetings were led by ML (patient and public involvement adult researcher (PPIA) 1). Analysis was performed by ML, checked by EM, and discussed with EV. ML made the first draft of the article, which was subsequently discussed within the team and reviewed by EM and EV. All authors read and approved the final manuscript.

**Funding** The PPI study itself was a low-cost project. Data collection for the main interview study was funded by the University of Groningen as part of the MD/PhD programme of Malou Luchtenberg. This funding body has no role in the design of the study and collection, analysis, interpretation of data or writing the manuscript.

**Competing interests** None declared.

**Patient and public involvement** Patients and/or the public were involved in the design, or conduct, or reporting, or dissemination plans of this research. Refer to the Methods section for further details.

**Patient consent for publication** Not required.

**Ethics approval** The Medical Ethics Review Board of the University Medical Center Groningen concluded that neither this study (M18.2334032, 24 July 2018) nor our larger study (M16.192386, 10 May 2016) fall within the scope of the Dutch Medical Research Involving Human Subjects Act.

**Provenance and peer review** Not commissioned; externally peer-reviewed.

**Data availability statement** No data are available. The original data set with audiotaped and videotaped interviews is not available for the public on account of privacy concerns.

**ORCID iD**
Malou L Luchtenberg http://orcid.org/0000-0002-9459-772X

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
