## [Reviewer comments · BMJ Open]

ARTICLE DETAILS

TITLE (PROVISIONAL)	'I actually felt like I was a researcher myself.' On Involving Children in the Analysis of Qualitative Paediatric Research in the Netherlands
AUTHORS	Luchtenberg, Malou; Maeckelberghe, Els; Verhagen, Eduard

VERSION 1 – REVIEW

REVIEWER	Jan Piasecki Department of Philosophy and Bioethics Faculty of Health Sciences Jagiellonian University Medical College
REVIEW RETURNED	30-Oct-2019

GENERAL COMMENTS	Reviewer's report Title: "I actually felt like I was a researcher myself": Involving Children as Co-Researchers in Analysis of Qualitative Paediatric Research Reviewer's recommendation: minor revisions before publication This paper is a report from a qualitative project on involving children as co-researchers in qualitative research concerning children participation in biomedical research. The project consisted of two phases. In the phase one children between 10-14 years old took part in a qualitative analysis of videotaped interviews (with children participating in biomedical research). This phase helped the adult researchers to determine shorter video clips that were analyzed in part two. In part one a child co-researcher worked with an adult researcher. In the phase-two video clips related to 2 themes identified in the previous phase were further analyzed in group meetings with children-co-researchers. The results of the research project is that: research with children as co-researchers is feasible and that adult researchers can get important insights from children. The recommendation for future projects like this one is that data should be presented in videos, not in transcripts. In my opinion the paper presents important and at the same time not sufficiently studied issue of children contribution to science, as co researchers. Comments and recommendations: 1. The authors write in their very first sentence that "Paediatric research is important to provide children with the best possible health care". (p. 3, line 4) I could not agree more. However I have not found in the paper a direct and explicit connection between the results of the research project and its possible impact on better health care with children. Further the authors quote one of the adult researchers saying: "I noticed that as an adult you tend to see things as obvious, for example which subtheme goes with a main theme, while children seem to have multiple other potential interpretations, and they are able to discuss those interpretation".
---

	But this statement seems to be ungrounded, because the authors do not give any concrete example: what exactly is the children co-researchers contribution, what they realized that the adult researchers did not manage to grasp (subtheme, main theme remark is too abstract). I guess that this might be topic of the next paper, however here at least some more tangible examples should be given. Recommendation: The authors are suggested to give at least one example: that clearly shows why it is worthy , form the standpoint of generating new knowledge and insights, to involve children as co-researchers. And moreover, how their involvement can change the health care (again concrete example). 2. This is a research (R1) on research with children as co-researchers (R2), the adults and children together do research on children involved in research (R3). Moreover research (R2) has two phases. This especially at the beginning of the paper is a bit complicated and seems not clear. Recommendation: I would suggest some figure or diagram that illustrate this complexity that it would be clearer. 3. The authors wrote that the involved children researchers gave their informed consent for participation in research on research with children (p. 4). But they do not explain, why they wanted to obtain informed consent, instead of assent that is usually taken from children. In my opinion the character of the project perfectly justifies obtaining consent, instead of assent (children are treated as equal co-researchers, whose opinions are important). However this justification is missing in the paper. Recommendation: I would suggest to add a few sentences, why consent was more appropriate.
--	---

REVIEWER	Chris A. Rees Boston Children's Hospital, Harvard Medical School
REVIEW RETURNED	23-Dec-2019

GENERAL COMMENTS	The authors report a reportedly new study approach that includes children 10-14 years of age as co-researchers to provide feedback on the way qualitative research involving children can be analyzed. This was a descriptive study within a study in what appears to be a single site in the Netherlands, though this detail is not clear. They describe the child's and adolescent's voice in research participation. The strength of this study is the exploration of an area, children's and adolescent's voice in research, that has not been well described, to my knowledge. Despite the article's strengths, there are several weaknesses that this reviewer thinks should be addressed. These weaknesses are outlined below in each section. The largest weaknesses are the lack of describing the larger context of the larger study and the rigor of the qualitative approach. Introduction: -Overall the introduction reads well. However, the final paragraph of the Introduction needs more information to convince the reader that this is an important and novel approach. For instance, how is "effective" defined? Effective at what? -The sentence, "In this paper we reflected on the process from the perspective of children and adults." is somewhat vague and merits further fleshing out. Methods:
---

	-It seems like the larger context of this study is lacking. The paper would be strengthened by describing what the objective of the larger study is. The details are there that would allow for reproducing the study, but the larger context is lacking. -It should be clearly stated what the study setting is. Meaning, was this a single-center study? Was this done at a single school? Please clarify this large and important piece of the study. -Please state what the purpose of the sessions was. -The actual analysis is lacking detail and rigor. How was the sampling done to account for diverse opinions? How did the authors attempt to establish credibility to their findings? Did the authors reach theme saturation? These are basic tenets of qualitative research that must be satisfied or at least accounted for. Results: -Basic information like participant demographics are lacking. It is imperative to include these to get a sense of how large the actual study was. -It would be helpful to know what the main themes from the analysis were. This could be stated up front. Themes are different from responses to each of the questions, which is how the Results seem to be organized currently. -Table 2 needs to specify if these are representative quotes or the authors summation of the qualitative findings. Particularly, the addition of exclamation marks if this is indeed summative is strange. -Page 7, lines 44-48. The authors need to provide supporting evidence for this claim. -The quotes included in the text seem to lack context. Adding simple words like, "one such representative quote was:" followed by the quote. -My personal preference is to include age and sex of the respondent for representative quotes as opposed to coding like PPI07, for example. Discussion: -The paragraph on Two-phase approach lacks comparison and contrast to existing literature as well as broader contextualization to help the reader understand how this work will lay the foundation for future work involving children in the research process as researchers, and not just respondents. Conclusions: -The line "The two-phase approach has the potential to prevent unrealistic interpretation of children's voices by adult researchers because it limits preselection of data by adults." is perhaps the most important line of the paper. I would suggest including this in the Abstract as well. -Lastly, are any of the authors these co-researchers? Just curious as authorship is the ultimate way of showing involvement. I am not saying the authors need to include the adolescents in the author byline, just something to note.
--	---

REVIEWER	Megan McHenry Indiana University, USA
REVIEW RETURNED	31-Dec-2019

GENERAL COMMENTS

This manuscript was focused on understanding the process and experience by which researchers might involve children in analyzing qualitative research. I appreciate the researchers' efforts to evaluate the process, while having the overall process aid in another study's analysis. I believe evaluating processes within a study can help others understand what is should be considered when trying new methods. However, I feel that as it is presented, there is not enough specific data that really helps me understand whether or not involving children is actually helpful or not helpful to the overall analysis. It is necessary to have a framework upon which to evaluate the child participant's contributions and perspectives on the research project. Also, much effort was used to describe the child participants within the study, however, there are perspectives and insights from the adult researchers, and it is unclear who those adult researchers are.

Another key concern that I had was that the authors explained that they did not want to train the children beforehand, "because [they] did not want them to become 'little adult researchers.'" However, the adults chose which themes identified in Phase 1 would be discussed in Phase 2, while only showing the participants a selected number of quotes/video. This seems to also introduce some bias. It is hard to ask anyone, adult or child, to add succinct contributions to large data sets without having any training or knowledge of how to do it. It will inevitably result in time and resources that may be better utilized. Inherently, I believe that without a strong framework to evaluate the comments and thoughts of the children regarding the data, one is only getting a superficial understanding of what they're contributing to the process. Because this study is separate from the manuscript where the authors describe the actual comments of the children, it's hard to gauge whether they made any impact or if it was just an interesting exercise in empowering young individuals to participate in research. I think combining these two studies would make for a more interesting manuscript

Other notes: Certain abbreviations were used, such as PPIA1 and PPIA2, which were not spelled out. Table 1 outlines the feedback questions asked from the participants, but there does not appear to be a clear summary of the answers to those questions within the results. Furthermore, as it is written now, it seems as though there was interpretation by the authors included within the results. For example, on Page 10, like 8-11, the results state one participant mentioned whether he would participate in the groups again would depend on whether he had time to take part. It goes on to describe this person was the only one in secondary school and so he had more homework, but as it's written, its just the assumption of the authors, not the statement of the actual child. Additionally, on Line 14 where one child acknowledged that participating might not suit everyone, there was no further follow up to elaborate on that answer. I think understanding why someone in this very small group might speak up and say some people might not like participating would be an important idea that should be explored further.

Ultimately, I believe that the results are interesting. However, with the current data presented, I think this manuscript would be stronger if it was combined with the actual contributions that the children made to the study they were analyzing.

VERSION 1 – AUTHOR RESPONSE

Reviewers' comment	Authors' response	Page numbers (in clean copy)
REVIEWER 1		
1. The authors write in their very first sentence that “Paediatric research is important to provide children with the best possible health care”. (p. 3, line 4) I could not agree more. However I have not found in the paper a direct and explicit connection between the results of the research project and its possible impact on better health care with children. Further the authors quote one of the adult researchers saying: “I noticed that as an adult you tend to see things as obvious, for example which subtheme goes with a main theme, while children seem to have multiple other potential interpretations, and they are able to discuss those interpretation”. But this statement seems to be ungrounded, because the authors do not give any concrete example: what exactly is the children co-researchers contribution, what they realized that the adult researchers did not manage to grasp (subtheme, main theme remark is too abstract). I guess that this might be topic of the next paper, however here at least some more tangible examples should be given. Recommendation: The authors are suggested to give at least one example: that clearly shows why it is worthy , form the standpoint of generating new knowledge and	Ad 1. Thank you for your comment. We agree that the direct link between the results and improving health was not formulated clearly enough. This is how we believe our results link to the statement that research is important: To provide children with good health care, we need research. However, children are seen as vulnerable group and professionals are sometimes reluctant to invite them to participate. Therefore, it is important to know what children themselves value, and how research can be adjusted to their needs. This is why we conducted the larger qualitative interview study on children’s experiences. To properly analyze the results, we believe children should be involved as co-researchers because we want to take their perspective seriously. Conducting good research will ultimately improve health care for children. We have adjusted our introduction to make this clearer. The reviewer is right that some more context is helpful to get an idea of the impact (content of analysis) of the project, so we have added a more tangible example to illustrate this. It will be further elaborated on in our next article.	3

insights, to involve children as co-researchers. And moreover, how their involvement can change the health care (again concrete example).		12
2. This is a research (R1) on research with children as co-researchers (R2), the adults and children together do research on children involved in research (R3). Moreover research (R2) has two phases. This especially at the beginning of the paper is a bit complicated and seems not clear. Recommendation: I would suggest some figure or diagram that illustrate this complexity that it would be clearer.	Ad 2. Thank you for the suggestion. We agree and have added a figure to illustrate this.	5 (text) Figure is attached separately
3. The authors wrote that the involved children researchers gave their informed consent for participation in research on research with children (p. 4). But they do not explain, why they wanted to obtain informed consent, instead of assent that is usually taken from children. In my opinion the character of the project perfectly justifies obtaining consent, instead of assent (children are treated as equal co-researchers, whose opinions are important). However this justification is missing in the paper. Recommendation: I would suggest to add a few sentences, why consent was more appropriate.	Ad 3. Thank you. We realize now that it should be explained in more detail. According to the Dutch national law, children between the age of 12 and 16 give consent in addition to their parents' consent. As the reviewer acknowledged, we have chosen to ask consent of all children (of all ages) because we treat them as equal co-researchers. Even though their contribution might be related or dependent (partly) on their ages, we believe all children have a unique contribution, and we value them all equally. We have added some sentences to explain this in the article.	4
REVIEWER 2		
Introduction: 1. Overall the introduction reads well. However, the final paragraph of the Introduction needs more information to convince the reader	Ad 1. We agree with the suggestion and have added further description of the importance.	3

that this is an important and novel approach. For instance, how is “effective” defined? Effective at what? 2. The sentence, “In this paper we reflected on the process from the perspective of children and adults.” is somewhat vague and merits further fleshing out.	Ad 2. We have adjusted the sentence to make it more clear.	3
Methods: 1. It seems like the larger context of this study is lacking. The paper would be strengthened by describing what the objective of the larger study is. The details are there that would allow for reproducing the study, but the larger context is lacking. 2. It should be clearly stated what the study setting is. Meaning, was this a single-center study? Was this done at a single school? Please clarify this large and important piece of the study. 3. Please state what the purpose of the sessions was. 4. The actual analysis is lacking detail and rigor. How was the sampling done to account for diverse opinions? How did the authors attempt to establish credibility to their findings? Did the authors reach theme saturation?	Ad 1. We have added a description of the overall goal of the original study in the introduction. In addition, we have added a table to provide more information about the original research and larger context. Ad 2. We agree, and have added a more detailed description of the study setting. Ad 3. The purpose of the sessions (individual + group meetings) is stated on p. 6 “The aim of the analysis process in both phases was to identify the main subjects present in the video through open, unstructured discussions.” Ad 4. Thank you for your feedback. We have added more detail about the methods we used.	3 5 4 6 6/7
Results: 1. Basic information like participant demographics are lacking. It is imperative to include these to get a sense of how large the actual study was. 2. It would be helpful to know what the main themes from the analysis were. This could be stated up front. Themes are different from	Ad 1. We have added participant characteristics (of both child -table 3- and adult -table 4- researchers). Ad 2. Thank you for your suggestion. We have added a description of the main themes in the beginning of the results section, and rearranged the description of our results, which are now organized based on the themes stated up front.	7/8 Table 3/4 9

responses to each of the questions, which is how the Results seem to be organized currently. 3. Table 2 needs to specify if these are representative quotes or the authors summation of the qualitative findings. Particularly, the addition of exclamation marks if this is indeed summative is strange. 4. Page 7, lines 44-48. The authors need to provide supporting evidence for this claim. 5. The quotes included in the text seem to lack context. Adding simple words like, "one such representative quote was:" followed by the quote. 6. My personal preference is to include age and sex of the respondent for representative quotes as opposed to coding like PPI07, for example.	Ad 3. Our aim with this table was to provide an overview of the overall findings. However, thanks to the reviewer we realize that it did not make the results clearer. Consequently, we decided to delete this table and provide only children's responses to the feedback form (described in the methods/table 1), which inform the themes described in the results section. Ad 4. Thank you for the feedback. We realize that this claim was stated too firmly. We have re-formulated it and provided more descriptive context to clarify and support this claim. Ad 5. Thank you for the suggestion. We have added more context to the quotes in the results section. Ad 6. We agree and have adjusted the details about the participants.	9 Table 5 9 Results section Results section
Discussion:	Ad 1. Thank you for the suggestion. We agree and have added existing literature to help the reader	14/15

1. The paragraph on Two-phase approach lacks comparison and contrast to existing literature as well as broader contextualization to help the reader understand how this work will lay the foundation for future work involving children in the research process as researchers, and not just respondents.	understand how this work could be seen in a broader context.	
Conclusions: 1. The line “The two-phase approach has the potential to prevent unrealistic interpretation of children’s voices by adult researchers because it limits preselection of data by adults.” is perhaps the most important line of the paper. I would suggest including this in the Abstract as well. 2. Lastly, are any of the authors these co-researchers? Just curious as authorship is the ultimate way of showing involvement. I am not saying the authors need to include the adolescents in the author byline, just something to note.	Ad 1. Thank you for the wonderful suggestion, we have added the sentence to our abstract. Ad 2. Thank you for the question. We believe it is very important to acknowledge the work which was done by our child co-researchers, which we see as equal collaborators in the analysis of the interviews. However, as this is a complex ‘meta-analysis’, we decided to not invite them in this article. (This is now added to the ‘acknowledgement section’.) In addition, the co-researchers were relatively young and do not have English as their native language. In our follow-up project with young people (aged 16-18) we are working on a shared article.	2
REVIEWER 3		
I believe evaluating processes within a study can help others understand what is should be considered when trying new methods. However, I feel that as it is presented, there is not enough specific data that really helps me understand whether or not involving children is actually helpful or not helpful to the overall analysis. It is necessary to have a framework upon which to evaluate the child participant’s contributions and perspectives on the research project. Also, much effort was used to describe the child participants within the study, however, there are perspectives	Thank you for these helpful considerations. We share these concerns and agree that a framework for analyzing the impact is helpful in order to describe and evaluate the participants’ contribution. However, given the fact that so little is known about how to involve children in research, we felt that filling this knowledge gap should be the first step. Because we want to provide a detailed description of how we performed this project, we decided to evaluate the impact in our next article. We agree that it is helpful to show some details on the impact to give the reader insight into what this impact could be. Therefore, we have added a concrete example (page 12). This will need to be further explored in the study on the impact of the co-researchers contribution on the actual analysis.	12

and insights from the adult researchers, and it is unclear who those adult researchers are.	We have also added more details about who the adult researchers are.	3 Table 4
Another key concern that I had was that the authors explained that they did not want to train the children beforehand, “because [they] did not want them to become ‘little adult researchers.’” However, the adults chose which themes identified in Phase 1 would be discussed in Phase 2, while only showing the participants a selected number of quotes/video. This seems to also introduce some bias. It is hard to ask anyone, adult or child, to add succinct contributions to large data sets without having any training or knowledge of how to do it. It will inevitably result in time and resources that may be better utilized. Inherently, I believe that without a strong framework to evaluate the comments and thoughts of the children regarding the data, one is only getting a superficial understanding of what they’re contributing to the process. Because this study is separate from the manuscript where the authors describe the actual comments of the children, it’s hard to gauge whether they made any impact or if it was just an interesting exercise in	Thank you for sharing our concerns. We believe limiting selection bias is one of the most important challenges in this field, as we do not want to have a tokenistic outcome. The preselection of themes by adults in the (promising) method of Best et al (described in introduction) was one of their limitations. Therefore, we aimed to explore how we could limit this by developing the 2-phase approach. In phase one, themes are identified by children (out of entire interviews). These themes are further explored in phase 2. Because this was an exploratory study, we have chosen to elaborate on one of the identified themes. Unfortunately, it was not feasible in this exploratory study to explore all main themes that were identified in phase 1. We agree that this would be desirable. In phase 2, we presented a selection of video fragments where interviewees were directly referring to that theme. Indirect referrals were excluded because in assessing that indirect referral lies a bias as well. We do agree with the reviewer that there will always be some bias, and we believe it is important to acknowledge this. This is why we stated this as	

empowering young individuals to participate in research. I think combining these two studies would make for a more interesting manuscript	study limitation at the end of the discussion section. We understand why the reviewer suggests to combine the answers to how this project was performed and evaluated, and the impact of the project (on the content of analysis). However, as explained above, in this new area we believe it is important to provide detailed evidence on both parts. This is why we reported on the 'how' first, and describe the 'impact' in more detail in a next article. We hope the reviewer could agree with us on this.	17
Other notes:  1. Certain abbreviations were used, such as PPIA1 and PPIA2, which were not spelled out. 2. Table 1 outlines the feedback questions asked from the participants, but there does not appear to be a clear summary of the answers to those questions within the results. 3. Furthermore, as it is written now, it seems as though there was interpretation by the authors included within the results For example, on Page 10, like 8-11, the results state one participant mentioned whether he would participate in the groups again would depend on whether he had time to take part. It goes on to describe this person was the only one in secondary school and so he had more homework, but as it's written, it's just the assumption of the authors, not the statement of the actual child. 4. Additionally, on Line 14 where one child acknowledged that participating might not suit everyone, there was no further follow up to elaborate on that answer. I think understanding why 	Ad 1. We agree and have spelled it out. We also changed the abbreviations into age and sex of the participants (as reviewer 2 suggested). Ad 2. We have added a table with the answers. (see also the answer to question 3 about the results section by reviewer 2) Ad 3. Thank you, we realize the wording does not represent what we tried to say. The example that the reviewer mentioned is an explanation of the child. We have changed the wording. Ad 4. We agree and have added a further exploration.	9 Table 5 10 10

someone in this very small group might speak up and say some people might not like participating would be an important idea that should be explored further.		
--	--	--

VERSION 2 – REVIEW

REVIEWER	Jan Piasecki Department of Philosophy and Bioethics, Faculty of Health Sciences, Jagiellonian University Medical College, Krakow, Poland
REVIEW RETURNED	03-Feb-2020

GENERAL COMMENTS	Reviewer's recommendation: to accept for publication, with possible minor improvements Summary of the paper after revision: This is a paper that aims at assessment of feasibility and value of cooperation with children as co-researchers in empirical, qualitative research concerning pediatric health. The problems has not been thoroughly explored, however it is methodologically and ethically important issue. Methodological aspect: we do not know how skewed is qualitative analysis of adult researchers and if they are able to capture what is important for children experiencing sickness. Ethical: if the adult's perspective is biased, we are not able to address problems that are important for children participating in biomedical research or undergoing medical treatment. In my opinion the authors addressed all important remarks and comments raised by reviewers. The current version presents the study in a more clear and transparent manner. The conclusions are predicated on the presented materials and results. Before publication I would suggests some minor improvements of the manuscript. Minor recommendations:  1. P. 3. Paragraph 1: I would suggest to move the first paragraph to the bottom of the section: Introduction. The reason is simple: a paper, from a reader perspective, should start from explaining the title and the main problem tackled further. The authors do that in the second paragraph of this section. Now the first paragraph is justification of the topic and explanation of its meaning and value – therefore it seems, at least to me, it should be move the bottom of the Introduction section. 2. P. 3. Line: 24-25: it: “we wanted to involve children in our analysis to strengthen our analysis”; it should be: “we wanted to involve children to strengthen our analysis”. 3. P. 5. Caption of the Table 1: Is: “Details of the larger interview study....”; should be: “Details of the larger interview study 1, see Fig. 1”. 4. P. 12: I would suggests to add some reference to psychological development literature: it's very important finding of the study, that children stick to more concrete things. And it seems also supported by the psychological finding about cognitive development.
---

REVIEWER	Chris A. Rees Boston Children's Hospital, Harvard Medical School
REVIEW RETURNED	02-Feb-2020

GENERAL COMMENTS	The authors have been very responsive to my original comments and suggestions. I appreciate the much-improved state of the manuscript. A few remaining comments below. Major Comments -I would appreciate a clear statement of the study design in the Methods. Yes, the authors now describe that this was a single-center study in the Netherlands, but they should clearly state in the Abstract methods that they conducted a qualitative study. Abstract -Minor word change suggestion, under Participants, should be “while taking part in medical research” not “with”. -Results, the sentence, “The extent to which they needed time and support in structuring varied.” is unclear and needs to be expanded upon and its meaning clearly stated. -Conclusions, first sentence. I don’t think “the two-phase approach is promising”. The structure of meeting with children in their homes and at their school is not novel. The novelty lies in having adolescents assist with theme identification in qualitative research. This should be clearly spelled out. Introduction -Perhaps a better description of the new method put forth by Best et al. would be helpful. Participatory theme elicitation is unclear as it currently reads. This description will also help the reader understand the novelty of the study at hand. -The final paragraph could be restructured. The final sentence seems to dangle and is incomplete. The second to last sentence, beginning with “Measuring the impact of a PPI process...” should be moved to an earlier point in the Introduction, not when the authors are proposing what they do in this manuscript. Methods -Is it an exploratory study or an exploratory analysis of qualitative data from children as co-researchers on collecting qualitative data? -Under “Data characteristics”, the description of Figure 1 is very unclear. Why refer to each study by study 1, 2, and 3 here? Also, the authors keep saying they are reporting a two-phase study, but the description says this study is Study 3. This language needs to be clarified. The use of “study 1, study 2, study 3” confuses more than clarifies. -Table 1 could be supplemental material instead of a table in the Methods. Though I defer to the editor’s decision on this. -The section Patient and public involvement adds little to the Methods of the paper. This should be in the final paragraph of the Introduction. -Was everything done in English? Results -14 children were included, but how many children were approached? -I appreciate the authors’ inclusion of Table 5 but as it currently stands, it is a little confusing to read through. Can the authors remove the question and summarize the content? Can they add
---

	columns instead of rows in the table for each response? Percentages for responses should be included, even though the sample size is very small. Also, the way the results are presented in Table 5 make it seem like there were choices provided for the child co-investigators but the methods read more like a qualitative approach. -Page 12, representative quotes at the bottom of the page. Would recommend removing these. They add little to the Results. -Page 13 under Reflection on time investment, what does adequate mean in reference to time investment? Do the authors mean appropriate? Discussion -Define INVOLVE on page 16. -Use of "win-win" situation is a little colloquial. Recommend changing. Lastly, there are many grammatical errors throughout this draft. The English should be cross-checked by a native English speaker or closely read by the authors to remove all errors.
--	---

REVIEWER	Megan McHenry Indiana University, USA
REVIEW RETURNED	11-Feb-2020

GENERAL COMMENTS	General: I applaud the authors for the work that they did in addressing reviewer comments. I believe I have a better sense of the overall project now because of the increased level of detail. However, with this understanding and re-read of the article, I believe that the work described would be better framed within a methods paper, rather than a study. As it is now, there is no clear objective within the introduction- just that this new approach was going to be tried and the authors would see how it goes. There weren't clear hypotheses or outcomes that were being evaluated. The authors say they want to "reflect on how we involved children in data analysis," however the themes talk more about how what the children thought about participating, and how really what the output is. The authors added in a few examples of statements the children made, but it still doesn't really address a stated objective. Additionally, I can't imagine the difficulty of writing in a second language, however the phrasing and sentence structure are not to the level needed for academic writing in English. Likely some of the eloquence of the original thought process is lost when having to translate it into English. Abstract: The first phrase in the Setting Section is not a complete sentence. Please list the number of children within the abstract and describe how the transcripts/audio/groups were analyzed within the abstract Introduction: - Page 4, line 8. "This results in reluctance to invite children" either requires punctuation, or preferably, needs further description, as in "This results in reluctance to include children in research." - Please review this section for appropriate use of tense and articles. Methods: - We typically call informed consent forms for children "assents"
---

	- Page 6, Lines 15-53 are quite confusing and take some time to understand. I appreciate that a figure was made to help me understand. With this understanding, I'm unsure if this project is appropriate to be deemed as a study, especially as I'm having a difficult time identifying the clear objective and outcome the authors were looking for. - I'm grateful that the authors wrote in what qualitative method they used, but it is still unclear to me how they used a framework approach from their description. It is also unclear if there were transcripts of the sessions that were coded or if it was something this individual summarized from the experiences of being the groups, which is at risk for bias. Results: -It's not clear to me how the themes were intended to answer the "how" of "how to involve children in analysis" The only theme that seemed to match with this was the time investment. Perhaps if it was intended to determine acceptability of participating, some of the themes would have matched better. - I appreciate the re-wording of Page 11, Lines 44-51. This is more clear. Discussion: -On page 13, Lines 24, the authors describe how the children's findings from the data were quite concrete, and they mostly described exactly what was said by the participant, rather than generalizing or thinking of them within a more integrated context. However, that is what adolescents do- developmentally, thinking critically is a bit harder for them and they still need practice at it with new tasks. I believe that's worth mentioning and state whether or not that's beneficial to the outcomes (it's actually much more aligned with what happens during line-by-line coding, which wasn't described here.) - Time investment was a critical theme to what I believe you were wanting to understand with this project, but there wasn't a lot of details regarding the amount of time it took to get these groups ready, etc. I believe I only saw times for the actually group/interview time.
--	---

VERSION 2 – AUTHOR RESPONSE

Reviewers' comments	Authors' responses	Page numbers (in clean copy)
REVIEWER 1		
In my opinion the authors addressed all important remarks and comments raised by reviewers. The current version presents the study in a more clear and transparent manner. The conclusions are predicated on the presented materials and results. Before publication I would suggest some minor improvements of the manuscript.	Thank you for your kind words.	

1. P. 3. Paragraph 1: I would suggest moving the first paragraph to the bottom of the section: Introduction. The reason is simple: a paper, from a reader perspective, should start from explaining the title and the main problem tackled further. The authors do that in the second paragraph of this section. Now the first paragraph is justification of the topic and explanation of its meaning and value – therefore it seems, at least to me, it should be moved to the bottom of the Introduction section.	1. Thank you for the suggestion. We welcome the advice and rearranged our introduction.	3
2. P. 3. Line: 24-25: it: “we wanted to involve children in our analysis to strengthen our analysis”; it should be: “we wanted to involve children to strengthen our analysis”.	2. Thank you. We have changed this sentence. The wording was changed slightly when we rearranged the introduction (see the previous comment)	3
3. P. 5. Caption of the Table 1: Is: “Details of the larger interview study...”; should be: “Details of the larger interview study 1, see Fig. 1”.	3. Following the other reviewers’ advice, we decided to delete the figure because it did not appear to make matters clearer.	-
4. P. 12: I would suggest adding some reference to psychological development literature: it’s very important finding of the study that children stick to more concrete things. And it seems also supported by the psychological finding about cognitive development.	4. Thank you for this most relevant suggestion. We agree that it adds value to our discussion.	17
REVIEWER 2		
The authors have been very responsive to my original comments and suggestions. I appreciate the much-improved state of the manuscript. A few remaining comments below. Major Comments 1.-I would appreciate a clear statement of the study design in the Methods. Yes, the authors now describe that this was a single-centre study in the Netherlands, but they should clearly state in the Abstract methods that they conducted a qualitative study.	Thank you for acknowledging our efforts to improve the manuscript. 1. Thank you for your advice. We included this added information to the settings section of the abstract.	2
Abstract 2. Minor word change suggestion, under Participants, should be “while taking part in medical research” not “with”. 3. -Results, the sentence, “The extent to which they needed time and support in structuring varied.” is unclear and needs to be expanded upon and its meaning clearly stated.	Thank you for the recommendations. 2. ‘Participants’: We changed the wording. 3. ‘Results’: We clarified this point.	2

4. -Conclusions, first sentence. I don't think "the two-phase approach is promising". The structure of meeting with children in their homes and at their school is not novel. The novelty lies in having adolescents assist with theme identification in qualitative research. This should be clearly spelled out.	4. 'Conclusion': thank you for this comment. On re-reading the sentence we realized that it is vague. We restructured the paragraph.	
Introduction 5. -Perhaps a better description of the new method put forth by Best et al. would be helpful. Participatory theme elicitation is unclear as it currently reads. This description will also help the reader understand the novelty of the study at hand. 6. The final paragraph could be restructured. The final sentence seems to dangle and is incomplete. The second to last sentence, beginning with "Measuring the impact of a PPI process..." should be moved to an earlier point in the Introduction, not when the authors are proposing what they do in this manuscript.	5. Thank you for the suggestion. We agree with it and added a description. 6. Thank you for this comment. We rearranged the paragraph, thereby also considering the feedback provided by Reviewer 1.	3
Methods 7.-Is it an exploratory study or an exploratory analysis of qualitative data from children as co-researchers on collecting qualitative data? 8. -Under "Data characteristics", the description of Figure 1 is very unclear. Why refer to each study by study 1, 2, and 3 here? Also, the authors keep saying they are reporting a two-phase study, but the description says this study is Study 3. This language needs to be clarified. The use of "study 1, study 2, study 3" confuses more than clarifies. 9. Table 1 could be supplemental material instead of a table in the Methods. Though I defer to the editor's decision on this. 10. -The section Patient and public involvement adds little to the Methods of the paper. This should be in the final paragraph of the Introduction. 11. -Was everything done in English?	7.Thank you. We realize that it was not clear. We have changed the wording. 8. We agree with your comment and after some deliberation decided to remove the figure altogether. 9. Thank you for the suggestion. Perhaps we misunderstood the instructions. We found the following: "Tables ... should be in Word format and placed in the main text where the table is first cited." (https://authors.bmj.com/writing-and-formatting/formatting-your-paper/) Nevertheless, should the editor prefer the table to be presented as a supplementary file, we shall of course comply. 10. Thank you for thinking along with us. We have followed your advice.	5 5-6

	11. The meetings were all in Dutch. We added this to our recruitment/sampling description, because it follows the inclusion criterion of being fluent in Dutch.	3-4 4
Results 12. -14 children were included, but how many children were approached? 13. -I appreciate the authors' inclusion of Table 5 but as it currently stands, it is a little confusing to read through. Can the authors remove the question and summarize the content? Can they add columns instead of rows in the table for each response? Percentages for responses should be included, even though the sample size is very small. Also, the way the results are presented in Table 5 make it seem like there were choices provided for the child co-investigators, but the methods read more like a qualitative approach. 14. -Page 12, representative quotes at the bottom of the page. Would recommend removing these. They add little to the Results. 15. -Page 13 under Reflection on time investment, what does adequate mean in reference to time investment? Do the authors mean appropriate?	12. In Phase 2, ten out of fifteen pupils volunteered (one of them was ill during the meeting). This is stated in the methods section under recruitment and sampling. Because we used different approaches for recruitment, including social media, we cannot provide information on how many children read our call for co-researchers. We did, however, add information on how many children requested the information pack in Phase one. 13. Thank you for expressing your concern. We looked at the table closely and rearranged it. We tried several structures to see which fitted best, including adding columns instead of rows. In the end we chose a similar layout to the one in Table 3. Currently, it is structured in line with the presentation of the rest of our data. Questions were removed as you suggested, but we did add a reference to which question on the form, which is presented in Table 2, is addressed. The table summarizes the answers of the respondents, which indeed are qualitative data. Only two sections show answers to multiple choice questions, which are now indicated with ***. We also added percentages.	4 9-10

	14. Thank you. We agree and removed this part. 15. Yes, we meant appropriate. Thank you for pointing this out.	-
		14
Discussion 16. -Define INVOLVE on page 16. 17. -Use of “win-win” situation is a little colloquial. Recommend changing.	16. We have added a description. 17. Thank you, we changed the wording.	17
Lastly, there are many grammatical errors throughout this draft. The English should be cross-checked by a native English speaker or closely read by the authors to remove all errors.	Thank you for mentioning this. After the current revision, we sent the manuscript to a professional/native speaker who edited the English text.	
REVIEWER 3		
I applaud the authors for the work that they did in addressing reviewer comments. I believe I have a better sense of the overall project now because of the increased level of detail. However, with this understanding and re-read of the article, I believe that the work described would be better framed	Thank you for acknowledging the improvement of our manuscript.	

within a methods paper, rather than a study. As it is now, there is no clear objective within the introduction- just that this new approach was going to be tried and the authors would see how it goes. There weren't clear hypotheses or outcomes that were being evaluated. The authors say they want to "reflect on how we involved children in data analysis," however the themes talk more about how what the children thought about participating, and how really what the output is. The authors added in a few examples of statements the children made, but it still doesn't really address a stated objective. Additionally, I can't imagine the difficulty of writing in a second language, however the phrasing and sentence structure are not to the level needed for academic writing in English. Likely some of the eloquence of the original thought process is lost when having to translate it into English.	We welcome your feedback on our study. It made us re-think our objectives and hypothesis. We realized that our objective was not formulated accurately. We aimed to evaluate the feasibility of our two-phase approach, rather than testing how children could be involved, hypothesizing that it is feasible to involve children in data analysis, including theme identification from original data. In addition, we wanted to reflect on children's experiences during the involvement process, rather than reflecting on how they were involved. The comment on our English grammar was also given by one of the other reviewers. Therefore before resubmitting the revised manuscript, we asked a professional/native speaker to edit the English text.	
1. Abstract: The first phrase in the Setting Section is not a complete sentence. Please list the number of children within the abstract and describe how the transcripts/audio/groups were analysed within the abstract	1. Thank you, we clarified this point.	2
2. Introduction: - Page 4, line 8. "This results in reluctance to invite children" either requires punctuation, or preferably, needs further description, as in "This results in reluctance to include children in research." - Please review this section for appropriate use of tense and articles.	2. Thank you for your comment. We have rearranged the introduction paragraphs following the other reviewers' feedback. We deleted the sentence in question.	3
Methods: 3. We typically call informed consent forms for children "assents" 4. Page 6, Lines 15-53 are quite confusing and take some time to understand. I appreciate that a figure was made to help me understand. With this understanding, I'm unsure if this project is appropriate to be deemed as a study, especially as I'm having a difficult time identifying the clear	3. According to Dutch law, children aged between 12 and 16 can give informed consent in addition to parental consent. In this study, we asked all the children to give informed consent to acknowledge their equally valued contribution. This is	-

objective and outcome the authors were looking for. 5. I'm grateful that the authors wrote in what qualitative method they used, but it is still unclear to me how they used a framework approach from their description. It is also unclear if there were transcripts of the sessions that were coded or if it was something this individual summarized from the experiences of being the groups, which is at risk for bias.	why we used consent forms instead of assents. 4. On rereading this section, we agree that it is unclear and that the description and figure do not match the set-up of our study as presented here. We decided to remove the figure. 5. Thank you for sharing your concern. We used different types of data, viz. written feedback forms, audio tapes of the meetings, and field notes, which were coded and checked by several authors. Any disagreements were discussed. We have extended our description of the framework approach we used. There were no transcriptions because of the number of voices intermingling during the meeting, including co-researchers and video data that was presented. Nevertheless, this part of the data was still recorded and themes were derived from multiple datasets.	7
Results: 6. It's not clear to me how the themes were intended to answer the "how" of "how to involve children in analysis" The only theme that seemed to match with this was the time investment. Perhaps if it was intended to determine acceptability of participating, some of the themes would have matched better. 7. I appreciate the re-wording of Page 11, Lines 44-51. This is more clear.	6. Thank you for this comment. Please see our explanation of your first overall comment. In our opinion our adjustments to the results section now match with the objectives. We trust you agree with us on this point. 7. Thank you for your feedback.	-
Discussion: 8. On page 13, Lines 24, the authors describe how the children's findings from the data were quite concrete, and they mostly described exactly what was said by the participant, rather than generalizing or	8. Thank you for this advice. It was also noticed by the first reviewer and we	17

thinking of them within a more integrated context. However, that is what adolescents do- developmentally, thinking critically is a bit harder for them and they still need practice at it with new tasks. I believe that's worth mentioning and state whether or not that's beneficial to the outcomes (it's actually much more aligned with what happens during line-by-line coding, which wasn't described here.) 9. Time investment was a critical theme to what I believe you were wanting to understand with this project, but there wasn't a lot of details regarding the amount of time it took to get these groups ready, etc. I believe I only saw times for the actual group/interview time.	agree that the discussion is improved by mentioning it. In our evaluation of research output, that is the involvement process and result of qualitative data analysis of original study, we shall explore in detail how these developmental differences might support our findings, and how potential differences in development between adults, children, and young people may be useful in qualitative data analysis. 9. Thank you for noticing. We agree with your observation. In the methods section we added more details on the amount of time invested by the adult researchers in preparing the involvement sessions methods section.	- 6
--	---	-------------------

VERSION 3 – REVIEW

REVIEWER	Jan Piasecki Department of Philosophy and Bioethics, Faculty of Health Sciences, Jagiellonian University, Medical College
REVIEW RETURNED	06-May-2020

GENERAL COMMENTS	The goal of this paper is to assess feasibility and value of children input into qualitative data analysis, in empirical research concerning pediatric population and its health. The main merit of the paper is novelty of the problem, innovative and scrupulous methodological approach. After a series of revisions the paper readability and clarity was significantly improved. In my opinion the authors addressed all important remarks and comments raised by reviewers. I do not have further comments.
--

REVIEWER	Chris A Rees Boston Children's Hospital, Harvard Medical School
REVIEW RETURNED	05-May-2020

GENERAL COMMENTS	The authors have again been very responsive to both my feedback and that of other reviewers. My comments are minor at this point. Abstract: -I suggest removing, or clarifying the meaning of, the sentence in the Conclusions that reads "Preselection of data is thus limited".
---

	-Can remove the sentence that reads "We recommend using videos rather than transcripts" as the value of videos is implied in the final sentence of the Conclusions. Methods: -Page 13, line 12: suggest avoiding the word "chaos" as I doubt it was actual chaos. This would perhaps be better described as confusion due to competing voices, or something of that nature. Results: -Page 15, line 25: "of course" not "off course".
--	--

VERSION 3 – AUTHOR RESPONSE

Reviewers' comments	Authors' responses	Page numbers (in clean copy)
REVIEWER 1		
The main merit of the paper is novelty of the problem, innovative and scrupulous methodological approach. After a series of revisions the paper readability and clarity was significantly improved. In my opinion the authors addressed all important remarks and comments raised by reviewers. I do not have further comments.	We are very grateful for your feedback on our manuscript, and for expressing your acknowledgement with regard to how we addressed your concerns.	-
REVIEWER 2		
The authors have again been very responsive to both my feedback and that of other reviewers. My comments are minor at this point.	Thank you very much for acknowledging our efforts to improve the manuscript. We welcome your final points of concern to help us to further clarify the manuscript.	-
Abstract: -I suggest removing, or clarifying the meaning of, the sentence in the Conclusions that reads "Preselection of data is thus limited". -Can remove the sentence that reads "We recommend using videos rather than transcripts" as the value of videos is implied in the final sentence of the Conclusions.	These are helpful suggestions to improve clarity and avoid repetition in our abstract. We have reformulated the sentence about preselection, and removed the other sentence as you suggested.	2
Methods: -Page 13, line 12: suggest avoiding the word "chaos" as I doubt it was actual chaos. This would perhaps be better described as confusion due to competing voices, or something of that nature.	Indeed confusion better describes the situation. We happily changed this sentence.	12

Results: -Page 15, line 25: "of course" not "off course".	Thank you for pointing out this typo.	14
--	---------------------------------------	----